# Architectural Complexity Measures of Recurrent Neural Networks

**Saizheng Zhang**[1,*] **Yuhuai Wu**[2,*]**, Tong Che**[4]**, Zhouhan Lin**[1]**,**
**Roland Memisevic**[1,5]**, Ruslan Salakhutdinov**[3,5] **and Yoshua Bengio**[1,5]
[1]MILA, Université de Montréal, [2]University of Toronto, [3]Carnegie Mellon University,
[4]Institut des Hautes Études Scientifiques, France, [5]CIFAR

## Abstract

In this paper, we systematically analyze the connecting architectures of recurrent neural networks (RNNs). Our main contribution is twofold: first, we present a rigorous graph-theoretic framework describing the connecting architectures of RNNs in general. Second, we propose three architecture complexity measures of RNNs: (a) the **recurrent depth**, which captures the RNN's over-time nonlinear complexity, (b) the **feedforward depth**, which captures the local input-output non-linearity (similar to the "depth" in feedforward neural networks (FNNs)), and (c) the **recurrent skip coefficient** which captures how rapidly the information propagates over time. We rigorously prove each measure's existence and computability. Our experimental results show that RNNs might benefit from larger recurrent depth and feedforward depth. We further demonstrate that increasing recurrent skip coefficient offers performance boosts on long term dependency problems.

## 1 Introduction

Recurrent neural networks (RNNs) have been shown to achieve promising results on many difficult sequential learning problems [1, 2, 3, 4, 5]. There is also much work attempting to reveal the principles behind the challenges and successes of RNNs, including optimization issues [6, 7], gradient vanishing/exploding related problems [8, 9], analysing/designing new RNN transition functional units like LSTMs, GRUs and their variants [10, 11, 12, 13].

This paper focuses on another important theoretical aspect of RNNs: the connecting architecture. Ever since [14, 15] introduced different forms of "stacked RNNs", researchers have taken architecture design for granted and have paid less attention to the exploration of other connecting architectures. Some examples include [16, 1, 17] who explored the use of skip connections; [18] who pointed out the distinction of constructing a "deep" RNN from the view of the recurrent paths and the view of the input-to-hidden and hidden-to-output maps. However, they did not rigorously formalize the notion of "depth" and its implications in "deep" RNNs. Besides "deep" RNNs, there still remains a vastly unexplored field of connecting architectures. We argue that one barrier for better understanding the architectural complexity is the lack of a general definition of the connecting architecture. This forced previous researchers to mostly consider the simple cases while neglecting other possible connecting variations. Another barrier is the lack of quantitative measurements of the complexity of different RNN connecting architectures: even the concept of "depth" is not clear with current RNNs.

In this paper, we try to address these two barriers. We first introduce a general formulation of RNN connecting architectures, using a well-defined graph representation. Observing that the RNN undergoes multiple transformations not only feedforwardly (from input to output within a time step) but also recurrently (across multiple time steps), we carry out a quantitative analysis of the number of transformations in these two orthogonal directions, which results in the definitions of *recurrent depth*

---

and *feedforward depth*. These two depths can be viewed as general extensions of the work of [18]. We also explore a quantity called the *recurrent skip coefficient* which measures how quickly information propagates over time. This quantity is strongly related to vanishing/exploding gradient issues, and helps deal with long term dependency problems. Skip connections crossing different timescales have also been studied by [19, 15, 20, 21]. Instead of specific architecture design, we focus on analyzing the graph-theoretic properties of recurrent skip coefficients, revealing the fundamental difference between the regular skip connections and the ones which truly increase the recurrent skip coefficients. We rigorously prove each measure's existence and computability under the general framework.

We empirically evaluate models with different recurrent/feedforward depths and recurrent skip coefficients on various sequential modelling tasks. We also show that our experimental results further validate the usefulness of the proposed definitions.

## 2 General Formulations of RNN Connecting Architectures

RNNs are learning machines that recursively compute new states by applying transition functions to previous states and inputs. Its connecting architecture describes how information flows between different nodes. In this section, we formalize the concept of the connecting architecture by extending the traditional graph-based illustration to a more general definition with a *finite directed multigraph* and its *unfolded* version. Let us first define the notion of the *RNN cyclic graph* $\mathcal{G}_c$ that can be viewed as a cyclic graphical representation of RNNs. We attach "weights" to the edges in the cyclic graph $\mathcal{G}_c$ that represent time delay differences between the source and destination node in the unfolded graph.

**Definition 2.1.** *Let $\mathcal{G}_c = (V_c, E_c)$ be a weighted directed multigraph [2], in which $V_c = V_{\text{in}} \cup V_{\text{out}} \cup V_{\text{hid}}$ is a finite nonempty set of nodes, $E_c \subset V_c \times V_c \times \mathbb{Z}$ is a finite set of directed edges. Each $e = (u, v, \sigma) \in E_c$ denotes a directed weighted edge pointing from node $u$ to node $v$ with an integer weight $\sigma$. Each node $v \in V_c$ is labelled by an integer tuple $(i, p)$. $i \in \{0, 2, \cdots m - 1\}$ denotes the time index of the given node, where $m$ is the **period number** of the RNN, and $p \in S$, where $S$ is a finite set of node labels. We call the weighted directed multigraph $\mathcal{G}_c = (V_c, E_c)$ an RNN cyclic graph, if (1) For every edge $e = (u, v, \sigma) \in E_c$, let $i_u$ and $i_v$ denote the time index of node $u$ and $v$, then $\sigma = i_v - i_u + k \cdot m$ for some $k \in \mathbb{Z}$. (2) There exists at least one directed cycle [3] in $\mathcal{G}_c$. (3) For any closed walk $\omega$, the sum of all the $\sigma$ along $\omega$ is not zero.*

Condition (1) assures that we can get a periodic graph (repeating pattern) when unfolding the RNN through time. Condition (2) excludes feedforward neural networks in the definition by forcing to have at least one cycle in the cyclic graph. Condition (3) simply avoids cycles after unfolding. The cyclic representation can be seen as a time folded representation of RNNs, as shown in Figure 1(a). Given an RNN cyclic graph $\mathcal{G}_c$, we unfold $\mathcal{G}_c$ over time $t \in \mathbb{Z}$ by the following procedure:

**Definition 2.2** (**Unfolding**). *Given an RNN cyclic graph $\mathcal{G}_c = (V_c, E_c, \sigma)$, we define a new infinite set of nodes $V_{un} = \{(i + km, p) | (i, p) \in V, k \in \mathbb{Z}\}$. The new set of edges $E_{un} \in V_{un} \times V_{un}$ is constructed as follows: $((t, p), (t', p')) \in E_{un}$ if and only if there is an edge $e = ((i, p), (i', p'), \sigma) \in E$ such that $t' - t = \sigma$, and $t \equiv i(\mod m)$. The new directed graph $\mathcal{G}_{un} = (V_{un}, E_{un})$ is called the unfolding of $\mathcal{G}_c$. Any infinite directed graph that can be constructed from an RNN cyclic graph through unfolding is called an RNN unfolded graph.*

**Lemma 2.1.** *The unfolding $\mathcal{G}_{\text{un}}$ of any RNN cyclic graph $\mathcal{G}_c$ is a directed acyclic graph (DAG).*

Figure 1(a) shows an example of two graph representations $\mathcal{G}_{\text{un}}$ and $\mathcal{G}_c$ of a given RNN. Consider the edge from node $(1, 7)$ going to node $(0, 3)$ in $\mathcal{G}_c$. The fact that it has weight 1 indicates that the corresponding edge in $\mathcal{G}_{\text{un}}$ travels one time step, $((t+1, 7), (t+2, 3))$. Note that node $(0, 3)$ also has a loop with weight 2. This loop corresponds to the edge $((t, 3), (t+2, 3))$. The two kinds of graph representations we presented above have a one-to-one correspondence. Also, any graph structure $\theta$ on $\mathcal{G}_{\text{un}}$ is naturally mapped into a graph structure $\bar{\theta}$ on $\mathcal{G}_c$. Given an edge tuple $\bar{e} = (u, v, \sigma)$ in $\mathcal{G}_c$, $\sigma$ stands for the number of time steps crossed by $\bar{e}$'s covering edges in $E_{un}$, i.e., for every corresponding edge $e \in \mathcal{G}_{\text{un}}$, $e$ must start from some time index $t$ to $t + \sigma$. Hence $\sigma$ corresponds to the "time delay" associated with $e$. In addition, the *period number* $m$ in Definition 2.1 can be interpreted as the time length of the entire non-repeated recurrent structure in its unfolded RNN graph $\mathcal{G}_{\text{un}}$. In other words, shifting the $\mathcal{G}_{\text{un}}$ through time by $km$ time steps will result in a DAG which is

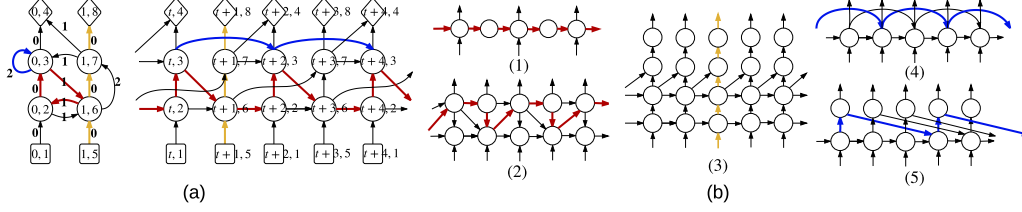

Figure 1: (a) An example of an RNN's $\mathcal{G}_c$ and $\mathcal{G}_{\mathrm{un}}$. $V_{\mathrm{in}}$ is denoted by square, $V_{\mathrm{hid}}$ is denoted by circle and $V_{\mathrm{out}}$ is denoted by diamond. In $\mathcal{G}_c$, the number on each edge is its corresponding $\sigma$. The longest path is colored in red. The longest input-output path is colored in yellow and the shortest path is colored blue. The value of three measures are $d_r = \frac{3}{2}$, $d_f = \frac{7}{2}$ and $s = 2$. (b) 5 more examples. (1) and (2) have $d_r = 2, \frac{3}{2}$, (3) has $d_f = 5$, (4) and (5) has $s = 2, \frac{3}{2}$.

identical to $\mathcal{G}_{\mathrm{un}}$, and $m$ is the smallest number that has such property for $\mathcal{G}_{\mathrm{un}}$. Most traditional RNNs have $m = 1$, while some special structures like *hierarchical* or *clockwork RNN* [15, 21] have $m > 1$. For example, Figure 1(a) shows that the period number of this specific RNN is 2.

The connecting architecture describes how information flows among RNN units. Assume $\bar{v} \in V_c$ is a node in $\mathcal{G}_c$, let $\mathrm{In}(\bar{v})$ denotes the set of incoming nodes of $\bar{v}$, $\mathrm{In}(\bar{v}) = \{\bar{u}|(\bar{u}, \bar{v}) \in E_c\}$. In the forward pass of the RNN, the transition function $F_{\bar{v}}$ takes outputs of nodes $\mathrm{In}(\bar{v})$ as inputs and computes a new output. For example, vanilla RNNs units with different activation functions, LSTMs and GRUs can all be viewed as units with specific transition functions. We now give the general definition of an RNN:

**Definition 2.3.** *An RNN is a tuple $(\mathcal{G}_c, \mathcal{G}_{\mathrm{un}}, \{F_{\bar{v}}\}_{\bar{v} \in V_c})$, in which $\mathcal{G}_{\mathrm{un}} = (V_{un}, E_{un})$ is the unfolding of RNN cyclic graph $\mathcal{G}_c$, and $\{F_{\bar{v}}\}_{\bar{v} \in V_c}$ is the set of transition functions. In the forward pass, for each hidden and output node $v \in V_{un}$, the transition function $F_{\bar{v}}$ takes all incoming nodes of $v$ as the input to compute the output.*

An RNN is *homogeneous* if all the hidden nodes share the same form of the transition function.

## 3  Measures of Architectural Complexity

In this section, we develop different measures of RNNs' architectural complexity, focusing mostly on the graph-theoretic properties of RNNs. To analyze an RNN solely from its architectural aspect, we make the mild assumption that the RNN is homogeneous. We further assume the RNN to be unidirectional. For a bidirectional RNN, it is more natural to measure the complexities of its unidirectional components.

### 3.1  Recurrent Depth

Unlike feedforward models where computations are done within one time frame, RNNs map inputs to outputs over multiple time steps. In some sense, an RNN undergoes transformations along both feedforward and recurrent dimensions. This fact suggests that we should investigate its architectural complexity from these two different perspectives. We first consider the recurrent perspective.

The conventional definition of depth is the *maximum* number of nonlinear transformations from inputs to outputs. Observe that a directed path in an unfolded graph representation $G_{un}$ corresponds to a sequence of nonlinear transformations. Given an unfolded RNN graph $G_{un}, \forall i, n \in \mathbb{Z}$, let $\mathfrak{D}_i(n)$ be the length of the *longest* path from any node at starting time $i$ to any node at time $i + n$. From the recurrent perspective, it is natural to investigate how $\mathfrak{D}_i(n)$ changes over time. Generally speaking, $\mathfrak{D}_i(n)$ increases as $n$ increases for all $i$. Such increase is caused by the recurrent structure of the RNN which keeps adding new nonlinearities over time. Since $\mathfrak{D}_i(n)$ approaches $\infty$ as $n$ approaches $\infty$,[4] to measure the complexity of $\mathfrak{D}_i(n)$, we consider its asymptotic behaviour, i.e., the limit of $\frac{\mathfrak{D}_i(n)}{n}$ as $n \to \infty$. Under a mild assumption, this limit exists. The following theorem prove such limit's computability and well-definedness:

**Theorem 3.2 (Recurrent Depth).** *Given an RNN and its two graph representation $\mathcal{G}_{\mathrm{un}}$ and $\mathcal{G}_c$, we denote $C(\mathcal{G}_c)$ to be the set of directed cycles in $\mathcal{G}_c$. For $\vartheta \in C(\mathcal{G}_c)$, let $l(\vartheta)$ denote the length of $\vartheta$*

*and $\sigma_s(\vartheta)$ denote the sum of edge weights $\sigma$ along $\vartheta$. Under a mild assumption[5],*

$$d_r = \lim_{n \to +\infty} \frac{\mathfrak{D}_i(n)}{n} = \max_{\vartheta \in C(\mathcal{G}_c)} \frac{l(\vartheta)}{\sigma_s(\vartheta)}. \tag{1}$$

More intuitively, $d_r$ is a measure of the average maximum number of nonlinear transformations per time step as $n$ gets large. Thus, we call it *recurrent depth*:

**Definition 3.1** (**Recurrent Depth**). *Given an RNN and its two graph representations $\mathcal{G}_{\mathrm{un}}$ and $\mathcal{G}_c$, we call $d_r$, defined in Eq.(1), the recurrent depth of the RNN.*

In Figure 1(a), one can easily verify that $\mathfrak{D}_t(1) = 5$, $\mathfrak{D}_t(2) = 6$, $\mathfrak{D}_t(3) = 8$, $\mathfrak{D}_t(4) = 9 \ldots$ Thus $\frac{\mathfrak{D}_t(1)}{1} = 5$, $\frac{\mathfrak{D}_t(2)}{2} = 3$, $\frac{\mathfrak{D}_t(3)}{3} = \frac{8}{3}$, $\frac{\mathfrak{D}_t(4)}{4} = \frac{9}{4}$ ...., which eventually converges to $\frac{3}{2}$ as $n \to \infty$. As $n$ increases, most parts of the longest path coincides with the path colored in red. As a result, $d_r$ coincides with the number of nodes the red path goes through per time step. Similarly in $\mathcal{G}_c$, observe that the red cycle achieves the maximum ($\frac{3}{2}$) in Eq.(1). Usually, one can directly calculate $d_r$ from $\mathcal{G}_{\mathrm{un}}$. It is easy to verify that *simple RNNs* and *stacked RNNs* share the same recurrent depth which is equal to 1. This reveals the fact that their nonlinearities increase at the same rate, which suggests that they will behave similarly in the long run. This fact is often neglected, since one would typically consider the number of layers as a measure of depth, and think of stacked RNNs as "deep" and simple RNNs as "shallow", even though their discrepancies are not due to recurrent depth (which regards time) but due to feedforward depth, defined next.

### 3.3 Feedforward Depth

Recurrent depth does not fully characterize the nature of nonlinearity of an RNN. As previous work suggests [3], stacked RNNs do outperform shallow ones with the same hidden size on problems where a more immediate input and output process is modeled. This is not surprising, since the growth rate of $\mathfrak{D}_i(n)$ only captures the number of nonlinear transformations in the time direction, not in the feedforward direction. The perspective of feedforward computation puts more emphasis on the specific paths connecting inputs to outputs. Given an RNN unfolded graph $G_{un}$, let $\mathfrak{D}_i^*(n)$ be the length of the longest path from any input node at time step $i$ to any output node at time step $i + n$. Clearly, when $n$ is small, the recurrent depth cannot serve as a good description for $\mathfrak{D}_i^*(n)$. In fact. it heavily depends on another quantity which we call *feedforward depth*. The following proposition guarantees the existence of such a quantity and demonstrates the role of both measures in quantifying the nonlinearity of an RNN.

**Proposition 3.3.1** (**Input-Output Length Least Upper Bound**). *Given an RNN with recurrent depth $d_r$, we denote $d_f = \sup_{i,n \in \mathbb{Z}} \mathfrak{D}_i^*(n) - n \cdot d_r$, the supremum $d_f$ exists and thus we have the following upper bound for $\mathfrak{D}_i^*(n)$:*

$$\mathfrak{D}_i^*(n) \leq n \cdot d_r + d_f.$$

The above upper bound explicitly shows the interplay between recurrent depth and feedforward depth: when $n$ is small, $\mathfrak{D}_i^*(n)$ is largely bounded by $d_f$; when $n$ is large, $d_r$ captures the nature of the bound ($\approx n \cdot d_r$). These two measures are equally important, as they separately capture the maximum number of nonlinear transformations of an RNN in the long run and in the short run.

**Definition 3.2.** (**Feedforward Depth**) *Given an RNN with recurrent depth $d_r$ and its two graph representations $\mathcal{G}_{\mathrm{un}}$ and $\mathcal{G}_c$, we call $d_f$, defined in Proposition 3.3.1, the feedforward depth[6] of the RNN.*

The following theorem proves $d_f$'s computability:

**Theorem 3.4** (**Feedforward Depth**). *Given an RNN and its two graph representations $\mathcal{G}_{\mathrm{un}}$ and $\mathcal{G}_c$, we denote $\xi(\mathcal{G}_c)$ the set of directed paths that start at an input node and end at an output node in $\mathcal{G}_c$. For $\gamma \in \xi(\mathcal{G}_c)$, denote $l(\gamma)$ the length and $\sigma_s(\gamma)$ the sum of $\sigma$ along $\gamma$. Then we have:*

$$d_f = \sup_{i,n \in \mathbb{Z}} \mathfrak{D}_i^*(n) - n \cdot d_r = \max_{\gamma \in \xi(\mathcal{G}_c)} l(\gamma) - \sigma_s(\gamma) \cdot d_r,$$

*where $m$ is the period number and $d_r$ is the recurrent depth of the RNN.*

For example, in Figure 1(a), one can easily verify that $d_f = \mathfrak{D}_t^*(0) = 3$. Most commonly, $d_f$ is the same as $\mathfrak{D}_t^*(0)$, i.e., the maximum length from an input to its current output.

## 3.5 Recurrent Skip Coefficient

Depth provides a measure of the complexity of the model. But such a measure is not sufficient to characterize behavior on long-term dependency tasks. In particular, since models with large recurrent depths have more nonlinearities through time, gradients can explode or vanish more easily. On the other hand, it is known that adding skip connections across multiple time steps may help improve the performance on long-term dependency problems [19, 20]. To measure such a "skipping" effect, we should instead pay attention to the length of the *shortest path* from time $i$ to time $i + n$. In $G_{un}$, $\forall i, n \in \mathbb{Z}$, let $\mathfrak{d}_i(n)$ be the length of the shortest path. Similar to the recurrent depth, we consider the growth rate of $\mathfrak{d}_i(n)$.

**Theorem 3.6** (**Recurrent Skip Coefficient**). *Given an RNN and its two graph representations $\mathcal{G}_{\mathrm{un}}$ and $\mathcal{G}_c$, under mild assumptions[7]*

$$j = \lim_{n \to +\infty} \frac{\mathfrak{d}_i(n)}{n} = \min_{\vartheta \in C(\mathcal{G}_c)} \frac{l(\vartheta)}{\sigma_s(\vartheta)}. \tag{2}$$

Since it is often the case that $j$ is smaller or equal to 1, it is more intuitive to consider its reciprocal.

**Definition 3.3.** (**Recurrent Skip Coefficient**)[8]. *Given an RNN and corresponding $\mathcal{G}_{\mathrm{un}}$ and $\mathcal{G}_c$, we define $s = \frac{1}{j}$, whose reciprocal is defined in Eq.(2), as the recurrent skip coefficient of the RNN.*

With a larger recurrent skip coefficient, the number of transformations per time step is smaller. As a result, the nodes in the RNN are more capable of "skipping" across the network, allowing unimpeded information flow across multiple time steps, thus alleviating the problem of learning long term dependencies. In particular, such effect is more prominent in the long run, due to the network's recurrent structure. Also note that not all types of skip connections can increase the recurrent skip coefficient. We will consider specific examples in our experimental results section.

# 4 Experiments and Results

In this section we conduct a series of experiments to investigate the following questions: (1) Is recurrent depth a trivial measure? (2) Can increasing depth yield performance improvements? (3) Can increasing the recurrent skip coefficient improve the performance on long term dependency tasks? (4) Does the recurrent skip coefficient suggest something more compared to simply adding skip connections? We show our evaluations on both $\tanh$ RNNs and LSTMs.

## 4.1 Tasks and Training Settings

**PennTreebank dataset**: We evaluate our models on character level language modelling using the PennTreebank dataset [22]. It contains 5059k characters for training, 396k for validation and 446k for test, and has a alphabet size of 50. We set each training sequence to have the length of 50. Quality of fit is evaluated by the bits-per-character (BPC) metric, which is $\log_2$ of perplexity.

**text8 dataset**: Another dataset used for character level language modelling is the text8 dataset[9], which contains $100M$ characters from Wikipedia with an alphabet size of 27. We follow the setting from [23] and each training sequence has length of 180.

**adding problem**: The adding problem (and the following copying memory problem) was introduced in [10]. For the adding problem, each input has two sequences with length of $T$ where the first sequence are numbers sampled from uniform[0, 1] and the second sequence are all zeros except two elements which indicates the position of the two elements in the first sequence that should be summed together. The output is the sum. We follow the most recent results and experimental settings in [24] (same for copying memory).

**copying memory problem**: Each input sequence has length of $T + 20$, where the first 10 values are random integers between 1 to 8. The model should remember them after $T$ steps. The rest of the sequence are all zeros, except for the last 11 entries in the sequence, which starts with 9 as a marker indicating that the model should begin to output its memorized values. The model is expected to give zero outputs at every time step except the last 10 entries, where it should generate (copy) the 10 values in the same order as it has seen at the beginning of the sequence. The goal is to minimize the average cross entropy of category predictions at each time step.

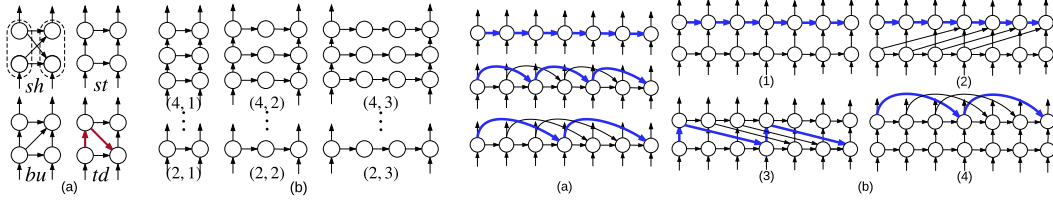

Figure 2: **Left**: (a) The architectures for $sh$, $st$, $bu$ and $td$, with their $(d_r, d_f)$ equal to $(1, 2)$, $(1, 3)$, $(1, 3)$ and $(2, 3)$, respectively. The longest path in $td$ are colored in red. (b) The 9 architectures denoted by their $(d_f, d_r)$ with $d_r = 1, 2, 3$ and $d_f = 2, 3, 4$. In both (a) and (b), we only plot hidden states at two adjacent time steps and the connections between them (the period number is 1). **Right**: (a) Various architectures that we consider in Section 4.4. From top to bottom are baseline $s = 1$, and $s = 2$, $s = 3$. (b) Proposed architectures that we consider in Section 4.5 where we take $k = 3$ as an example. The shortest paths in (a) and (b) that correspond to the recurrent skip coefficients are colored in blue.

| DATASET | MODELS\ARCHS | $sh$ | $st$ | $bu$ | $td$ |
|---------|--------------|------|------|------|------|
| PENNTREEBANK | tanh RNN | 1.54 | 1.59 | 1.54 | **1.49** |
| | tanh RNN-SMALL | 1.80 | 1.82 | 1.80 | **1.77** |
| TEXT8 | tanh RNN-LARGE | 1.69 | 1.67 | 1.64 | **1.59** |
| | LSTM-SMALL | 1.65 | 1.66 | 1.65 | **1.63** |
| | LSTM-LARGE | 1.52 | 1.53 | 1.52 | **1.49** |

| $d_f$\$d_r$ | $d_r = 1$ | $d_r = 2$ | $d_r = 3$ |
|-------------|-----------|-----------|-----------|
| $d_f = 2$ | 1.88 | 1.86 | 1.86 |
| $d_f = 3$ | 1.86 | **1.84** | 1.86 |
| $d_f = 4$ | 1.85 | 1.86 | 1.88 |

Table 1: **Left**: Test BPCs of $sh$, $st$, $bu$, $td$ for tanh RNNs and LSTMs. **Right**: Test BPCs of tanh RNNs with recurrent depth $d_r = 1, 2, 3$ and feedforward depth $d_f = 2, 3, 4$ respectively.

**sequential MNIST dataset**: Each MNIST image data is reshaped into a $784 \times 1$ sequence, turning the digit classification task into a sequence classification one with long-term dependencies [25, 24]. A slight modification of the dataset is to permute the image sequences by a fixed random order beforehand (permuted MNIST). Results in [25] have shown that both *tanh* RNNs and LSTMs did not achieve satisfying performance, which also highlights the difficulty of this task.

For all of our experiments we use *Adam* [26] for optimization, and conduct a grid search on the learning rate in $\{10^{-2}, 10^{-3}, 10^{-4}, 10^{-5}\}$. For tanh RNNs, the parameters are initialized with samples from a uniform distribution. For LSTM networks we adopt a similar initialization scheme, while the forget gate biases are chosen by the grid search on $\{-5, -3, -1, 0, 1, 3, 5\}$. We employ early stopping and the batch size was set to $50$.

## 4.2 Recurrent Depth is Non-trivial

To investigate the first question, we compare 4 similar connecting architectures: 1-layer (shallow) "$sh$", 2-layers stacked "$st$", 2-layers stacked with an extra bottom-up connection "$bu$", and 2-layers stacked with an extra top-down connection "$td$", as shown in Figure 2(a), left panel. Although the four architectures look quite similar, they have different recurrent depths: $sh$, $st$ and $bu$ have $d_r = 1$, while $td$ has $d_r = 2$. Note that the specific construction of the extra nonlinear transformations in $td$ is not conventional. Instead of simply adding intermediate layers in hidden-to-hidden connection, as reported in [18], more nonlinearities are gained by a recurrent flow from the first layer to the second layer and then back to the first layer at each time step (see the red path in Figure 2a, left panel).

We first evaluate our architectures using tanh RNN on PennTreebank, where $sh$ has hidden-layer size of 1600. Next, we evaluate four different models for text8 which are tanh RNN-small, tanh RNN-large, LSTM-small, LSTM large, where the model's $sh$ architecture has hidden-layer size of 512, 2048, 512, 1024 respectively. Given the architecture of the $sh$ model, we set the remaining three architectures to have the same number of parameters. Table 1, left panel, shows that the $td$ architecture outperforms all the other architectures for all the different models. Specifically, $td$ in tanh RNN achieves a test BPC of 1.49 on PennTreebank, which is comparable to the BPC of 1.48 reported in [27] using stabilization techniques. Similar improvements are shown for LSTMs, where $td$ architecture in LSTM-large achieves BPC of 1.49 on text8, outperforming the BPC of 1.54 reported in [23] with Multiplicative RNN (MRNN). It is also interesting to note the improvement we obtain when switching from *bu* to *td*. The only difference between these two architectures lies in changing the direction of one connection (see Figure 2(a)), which also increases the recurrent depth. Such a fundamental difference is by no means self-evident, but this result highlights the necessity of the concept of recurrent depth.

### 4.3 Comparing Depths

From the previous experiment, we found some evidence that with larger recurrent depth, the performance might improve. To further investigate various implications of depths, we carry out a systematic analysis for both recurrent depth $d_r$ and feedforward depth $d_f$ on text8 and sequential MNIST datasets. We build 9 models in total with $d_r = 1, 2, 3$ and $d_f = 2, 3, 4$, respectively (as shown in Figure 2(b)). We ensure that all the models have roughly the same number of parameters (e.g., the model with $d_r = 1$ and $d_f = 2$ has a hidden-layer size of 360).

Table 1, right panel, displays results on the text8 dataset. We observed that when fixing feedforward depth $d_f = 2, 3$ (or fixing recurrent depth $d_r = 1, 2$), increasing recurrent depth $d_r$ from 1 to 2 (or increasing feedforward depth $d_f$ from 2 to 3) does improve the model performance. The best test BPC is achieved by the architecture with $d_f = 3, d_r = 2$. This suggests that reasonably increasing $d_r$ and $d_f$ can aid in better capturing the over-time nonlinearity of the input sequence. However, for too large $d_r$ (or $d_f$) like $d_r = 3$ or $d_f = 4$, increasing $d_f$ (or $d_r$) only hurts models performance. This can potentially be attributed to the optimization issues when modelling large input-to-output dependencies (see Appendix B.4 for more details). With sequential MNIST dataset, we next examined the effects of $d_f$ and $d_r$ when modelling long term dependencies (more in Appendix B.4). In particular, we observed that increasing $d_f$ does not bring any improvement to the model performance, and increasing $d_r$ might even be detrimental for training. Indeed, it appears that $d_f$ only captures the local nonlinearity and has less effect on the long term prediction. This result seems to contradict previous claims [17] that stacked RNNs ($d_f > 1$, $d_r = 1$) could capture information in different time scales and would thus be more capable of dealing with learning long-term dependencies. On the other hand, a large $d_r$ indicates multiple transformations per time step, resulting in greater gradient vanishing/exploding issues [18], which suggests that $d_r$ should be neither too small nor too large.

### 4.4 Recurrent Skip Coefficients

To investigate whether increasing a recurrent skip coefficient $s$ improves model performance on long term dependency tasks, we compare models with increasing $s$ on the adding problem, the copying memory problem and the sequential MNIST problem (without/with permutation, denoted as MNIST and $p$MNIST). Our baseline model is the shallow architecture proposed in [25]. To increase the recurrent skip coefficient $s$, we add connections from time step $t$ to time step $t + k$ for some fixed integer $k$, shown in Figure 2(a), right panel. By using this specific construction, the recurrent skip coefficient increases from 1 (i.e., baseline) to $k$ and the new model with extra connection has 2 hidden matrices (one from $t$ to $t + 1$ and the other from $t$ to $t + k$).

For the adding problem, we follow the same setting as in [24]. We evaluate the baseline LSTM with 128 hidden units and an LSTM with $s = 30$ and 90 hidden units (roughly the same number of parameters as the baseline). The results are quite encouraging: as suggested in [24] baseline LSTM works well for input sequence lengths $T = 100, 200, 400$ but fails when $T = 750$. On the other hand, we observe that the LSTM with $s = 30$ learns perfectly when $T = 750$, and even if we increase $T$ to 1000, LSTM with $s = 30$ still works well and the loss reaches to zero.

For the copying memory problem, we use a single layer RNN with 724 hidden units as our basic model, and 512 hidden units with skip connections. So they have roughly the same number of parameters. Models with a higher recurrent skip coefficient outperform those without skip connections by a large margin. When $T = 200$, test set cross entropy (CE) of a basic model only yields 0.2409, but with $s = 40$ it is able to reach a test set cross entropy of 0.0975. When $T = 300$, a model with $s = 30$ yields a test set CE of 0.1328, while its baseline could only reach 0.2025. We varied the sequence length ($T$) and recurrent skip coefficient ($s$) in a wide range (where $T$ varies from 100 up to 300, and $s$ from 10 up to 50), and found that this kind of improvement persists.

For the sequential MNIST problem, the hidden-layer size of the baseline model is set to 90 and models with $s > 1$ have hidden-layer sizes of 64. The results in Table 2, top-left panel, show that tanh RNNs with recurrent skip coefficient $s$ larger than 1 could improve the model performance dramatically. Within a reasonable range of $s$, test accuracy increases quickly as $s$ becomes larger. We note that our model is the first tanh RNN model that achieves good performance on this task, even improving upon the method proposed in [25]. In addition, we also formally compare with the previous results reported in [25, 24], where our model (referred to as stanh) has a hidden-layer size of 95, which is about the same number of parameters as in the tanh model of [24]. Table 2, bottom-left panel, shows that our simple architecture improves upon the $u$RNN by 2.6% on $p$MNIST,

| stanh | s = 1 | s = 5 | s = 9 | s = 13 | s = 21 |
|---|---|---|---|---|---|
| MNIST | 34.9 | 46.9 | 74.9 | 85.4 | **87.8** |

| | s = 1 | s = 3 | s = 5 | s = 7 | s = 9 |
|---|---|---|---|---|---|
| *p*MNIST | 49.8 | 79.1 | 84.3 | **88.9** | 88.0 |

| LSTM | s = 1 | s = 3 | s = 5 | s = 7 | s = 9 |
|---|---|---|---|---|---|
| MNIST | 56.2 | **87.2** | 86.4 | 86.4 | 84.8 |

| | s = 1 | s = 3 | s = 4 | s = 5 | s = 6 |
|---|---|---|---|---|---|
| *p*MNIST | 28.5 | 25.0 | 60.8 | 62.2 | **65.9** |

| Model | MNIST | *p*MNIST |
|---|---|---|
| iRNN[25] | 97.0 | ≈82.0 |
| uRNN[24] | 95.1 | 91.4 |
| LSTM[24] | **98.2** | 88.0 |
| RNN(tanh)[25] | ≈35.0 | ≈35.0 |
| stanh(s = 21, 11) | 98.1 | **94.0** |

| Architecture, s | (1), 1 | (2), 1 | (3), $\frac{k}{2}$ | (4), $k$ |
|---|---|---|---|---|
| MNIST  k = 17 | 39.5 | 39.4 | 54.2 | **77.8** |
| k = 21 | 39.5 | 39.9 | 69.6 | **71.8** |
| *p*MNIST   k = 5 | 55.5 | 66.6 | 74.7 | **81.2** |
| k = 9 | 55.5 | 71.1 | 78.6 | **86.9** |

Table 2: Results for MNIST/*p*MNIST. **Top-left**: Test accuracies with different $s$ for tanh RNN. **Top-right**: Test accuracies with different $s$ for LSTM. **Bottom-left**: Compared to previous results. **Bottom-right**: Test accuracies for architectures (1), (2), (3) and (4) for tanh RNN.

and achieves almost the same performance as LSTM on the MNIST dataset with only 25% number of parameters [24]. Note that obtaining good performance on sequential MNIST requires a larger $s$ than that for *p*MNIST (see Appendix B.4 for more details). LSTMs also showed performance boost and much faster convergence speed when using larger $s$, as displayed in Table 2, top-right panel. LSTM with $s = 3$ already performs quite well and increasing $s$ did not result in any significant improvement, while in *p*MNIST, the performance gradually improves as $s$ increases from 4 to 6. We also observed that the LSTM network performed worse on permuted MNIST compared to a tanh RNN. Similar result was also reported in [25].

### 4.5 Recurrent Skip Coefficients vs. Skip Connections

We also investigated whether the recurrent skip coefficient can suggest something more than simply adding skip connections. We design 4 specific architectures shown in Figure 2(b), right panel. (1) is the baseline model with a 2-layer stacked architecture, while the other three models add extra skip connections in different ways. Note that **these extra skip connections all cross the same time length** $k$. In particular, (2) and (3) share quite similar architectures. However, ways in which the skip connections are allocated makes big differences on their recurrent skip coefficients: (2) has $s = 1$, (3) has $s = \frac{k}{2}$ and (4) has $s = k$. Therefore, even though (2), (3) and (4) all add extra skip connections, the fact that their recurrent skip coefficients are different might result in different performance.

We evaluated these architectures on the sequential MNIST and *p*MNIST datasets. The results show that differences in $s$ indeed cause big performance gaps regardless of the fact that they all have skip connections (see Table 2, bottom-right panel). Given the same $k$, the model with a larger $s$ performs better. In particular, model (3) is better than model (2) even though they only differ in the direction of the skip connections. It is interesting to see that for MNIST (unpermuted), the extra skip connection in model (2) (which does not really increase the recurrent skip coefficient) brings almost no benefits, as model (2) and model (1) have almost the same results. This observation highlights the following point: when addressing the long term dependency problems using skip connections, instead of only considering the time intervals crossed by the skip connection, one should also consider the model's recurrent skip coefficient, which can serve as a guide for introducing more powerful skip connections.

## 5 Conclusion

In this paper, we first introduced a general formulation of RNN architectures, which provides a solid framework for the architectural complexity analysis. We then proposed three architectural complexity measures: recurrent depth, feedforward depth, and recurrent skip coefficients capturing both short term and long term properties of RNNs. We also found empirical evidences that increasing recurrent depth and feedforward depth might yield performance improvements, increasing feedforward depth might not help on long term dependency tasks, while increasing the recurrent skip coefficient can largely improve performance on long term dependency tasks. These measures and results can provide guidance for the design of new recurrent architectures for particular learning tasks.

## Acknowledgments

The authors acknowledge the following agencies for funding and support: NSERC, Canada Research Chairs, CIFAR, Calcul Quebec, Compute Canada, Samsung, ONR Grant N000141310721, ONR Grant N000141512791 and IARPA Raytheon BBN Contract No. D11PC20071. The authors thank the developers of Theano [28] and Keras [29], and also thank Nicolas Ballas, Tim Cooijmans, Ryan Lowe, Mohammad Pezeshki, Roger Grosse and Alex Schwing for their insightful comments.

## Footnotes

[2]A directed multigraph is a directed graph that allows multiple directed edges connecting two nodes.

[3]A directed cycle is a closed walk with no repetitions of edges.

[4]Without loss of generality, we assume the unidirectional RNN approaches positive infinity.

[5]See a full treatment of the limit in general cases in Theorem A.1 and Proposition A.1.1 in Appendix.

[6]Conventionally, an architecture with depth 1 is a three-layer architecture containing one hidden layer. But in our definition, since it goes through two transformations, we count the depth as 2 instead of 1. This should be particularly noted with the concept of feedforward depth, which can be thought as the conventional depth plus 1.

[7]See Proposition A.3.1 in Appendix.

[8]One would find this definition very similar to the definition of the recurrent depth. Therefore, we refer readers to examples in Figure 1 for some illustrations.

[9]http://mattmahoney.net/dc/textdata.

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
