[Supplementary Material]

# A  Proofs

To show theorem 3.2, we first consider the most general case in which $d_r$ is defined (Theorem A.1). Then we discuss the mild assumptions under which we can reduce to the original limit (Proposition A.1.1). Additionally, we introduce some notations that will be used throughout the proof. If $v = (t, p) \in \mathcal{G}_{\mathrm{un}}$ is a node in the unfolded graph, it has a corresponding node in the folded graph, which is denoted by $\bar{v} = (\bar{t}, p)$.

**Theorem A.1.** *Given an RNN cyclic graph and its unfolded representation $(\mathcal{G}_c, \mathcal{G}_{\mathrm{un}})$, we denote $C(\mathcal{G}_c)$ the set of directed cycles in $\mathcal{G}_c$. For $\vartheta \in C(\mathcal{G}_c)$, denote $l(\vartheta)$ the length of $\vartheta$ and $\sigma_s(\vartheta)$ the sum of $\sigma$ along $\vartheta$. Write $d_i = \limsup_{k \to \infty} \frac{\mathfrak{D}_i(n)}{n}$.*[10] *we have :*

- *The quantity $d_i$ is periodic, in the sense that $d_{i+m} = d_i, \forall i \in \mathbb{N}$.*

- *Let $d_r = \max_i d_i$, then*

$$d_r = \max_{\vartheta \in C(\mathcal{G}_c)} \frac{l(\vartheta)}{\sigma_s(\vartheta)} \tag{3}$$

*Proof.* The first statement is easy to prove. Because of the periodicity of the graph, any path from time step $i$ to $i + n$ corresponds to an isomorphic path from time step $i + m$ to $i + m + n$. Passing to limit, and we can deduce the first statement.

Now we prove the second statement. Write $\vartheta_0 = \mathrm{argmax}_\vartheta \frac{l(\vartheta)}{\sigma_s(\vartheta)}$. First we prove that $d \geq \frac{l(\vartheta_0)}{\sigma_s(\vartheta_0)}$. Let $c_1 = (t_1, p_1) \in \mathcal{G}_{\mathrm{un}}$ be a node such that if we denote $\overline{c_1} = (\overline{t_1}, p_1)$ the image of $c_1$ on the cyclic graph, we have $\overline{c_1} \in \vartheta_0$. Consider the subsequence $S_0 = \left\{ \frac{\mathfrak{D}_{\overline{t_1}}(k\sigma_s(\vartheta_0))}{k\sigma_s(\vartheta_0)} \right\}_{k=1}^{\infty}$ of $\left\{ \frac{\mathfrak{D}_{\overline{t_1}}(n)}{n} \right\}_{n=1}^{\infty}$. From the definition of $\mathfrak{D}$ and the fact that $\vartheta_0$ is a directed circle, we have $\mathfrak{D}_{\overline{t_1}}(k\sigma_s(\vartheta_0)) \geq kl(\vartheta_0)$, by considering the path on $\mathcal{G}_{\mathrm{un}}$ corresponding to following $\vartheta_0$ $k$-times. So we have

$$d_r \geq \limsup_{k \to +\infty} \frac{\mathfrak{D}_i(n)}{n} \geq \limsup_{k \to +\infty} \frac{\mathfrak{D}_{\overline{t_1}}(k\sigma_s(\vartheta_0))}{k\sigma_s(\vartheta_0)} \geq \frac{kl(\vartheta_0)}{k\sigma_s(\vartheta_0)} = \frac{l(\vartheta_0)}{\sigma_s(\vartheta_0)}$$

Next we prove $d_r \leq \frac{l(\vartheta_0)}{\sigma_s(\vartheta_0)}$. It suffices to prove that, for any $\epsilon \geq 0$, there exists $N > 0$, such that for any path $\gamma : \{(t_0, p_0), (t_1, p_1), \cdots, (t_{n_\gamma}, p_{n_\gamma})\}$ with $t_{n_\gamma} - t_1 > N$, we have $\frac{n_\gamma}{t_{n_\gamma} - t_1} \leq \frac{l(\vartheta_0)}{\sigma_s(\vartheta_0)} + \epsilon$. We denote $\bar{\gamma}$ as the image of $\gamma$ on the cyclic graph. $\bar{\gamma}$ is a walk with repeated nodes and edges. Also, we assume there are in total $\Gamma$ nodes in cyclic graph $\mathcal{G}_c$.

We first decompose $\bar{\gamma}$ into a path and a set of directed cycles. More precisely, there is a path $\gamma_0$ and a sequence of directed cycles $C = C_1(\gamma), C_2(\gamma), \cdots, C_w(\gamma)$ on $\mathcal{G}_c$ such that:

- The starting and end nodes of $\gamma_0$ is the same as $\gamma$. (If $\gamma$ starts and ends at the same node, take $\gamma_0$ as empty.)

- The catenation of the sequences of directed edges $E(\gamma_0), E(C_1(\gamma)), E(C_2(\gamma)), \cdots, E(C_w(\gamma))$ is a permutation of the sequence of edges of $E(\gamma)$.

The existence of such a decomposition can be proved iteratively by removing directed cycles from $\gamma$. Namely, if $\gamma$ is not a paths, there must be some directed cycles $C'$ on $\gamma$. Removing $C'$ from $\gamma$, we can get a new walk $\gamma'$. Inductively apply this removal, we will finally get a (possibly empty) path and a sequence of directed cycles. For a directed path or loop $\gamma$, we write $D(\gamma)$ the distance between the ending node and starting node when travel through $\gamma$ once. We have

$$D(\gamma_0) := \overline{t_{n_\gamma}} - \overline{t_0} + \sum_{i=1}^{|\gamma_0|} \sigma(e_i)$$

where $e_i, i \in \{1, 2, \cdots, |\gamma_0|\}$ is all the edges of $\gamma_0$. $\bar{t}$ denotes the module of $t$: $t \equiv \bar{t} \pmod m$.

So we have:

$$|D(\gamma_0)| \le m + \Gamma \cdot \max_{e \in \mathcal{G}_c} \sigma(e) = M$$

For convenience, we denote $l_0, l_1, \cdots, l_w$ to be the length of path $\gamma_0$ and directed cycles $C_1(\gamma), C_2(\gamma), \cdots, C_w(\gamma)$. Obviously we have:

$$n_\gamma = \sum_{i=0}^{w} l_i$$

And also, we have

$$t_{n_\gamma} - t_1 = \sum_{i=1}^{w} \sigma_s(C_i) + D(\gamma_0)$$

So we have:

$$\frac{n_\gamma}{t_{n_\gamma} - t_1} = \frac{l_0}{t_{n_\gamma} - t_1} + \sum_{i=1}^{w} \frac{l_i}{t_{n_\gamma} - t_1} \le \frac{\Gamma}{N} + \sum_{i=1}^{w} \frac{l_i}{t_{n_\gamma} - t_1}$$

In which we have for all $i \in \{1, 2, \cdots, w\}$:

$$\frac{l_i}{t_{n_\gamma} - t_1} = \frac{l_i}{\sigma_s(C_i)} \cdot \frac{\sigma_s(C_i)}{t_{n_\gamma} - t_1} \le \frac{l(\vartheta_0)}{\sigma_s(\vartheta_0)} \frac{\sigma_s(C_i)}{t_{n_\gamma} - t_1}$$

So we have:

$$\sum_{i=1}^{w} \frac{l_i}{t_{n_\gamma} - t_1} \le \frac{l(\vartheta_0)}{\sigma_s(\vartheta_0)} \left[ 1 - \frac{D(\gamma_0)}{t_{n_\gamma} - t_1} \right] \le \frac{l(\vartheta_0)}{\sigma_s(\vartheta_0)} + \frac{M'}{N}$$

in which $M'$ and $\Gamma$ are constants depending only on the RNN $\mathcal{G}_c$.

Finally we have:

$$\frac{n_\gamma}{t_{n_\gamma} - t_1} \le \frac{l(\vartheta_0)}{\sigma_s(\vartheta_0)} + \frac{M' + \Gamma}{N}$$

take $N > \frac{M'+\Gamma}{\epsilon}$, we can prove the fact that $d_r \le \frac{l(\vartheta_0)}{\sigma_s(\vartheta_0)}$.

$\square$

**Proposition A.1.1.** Given an RNN and its two graph representations $\mathcal{G}_{un}$ and $\mathcal{G}_c$, if $\exists \vartheta \in C(\mathcal{G}_c)$ such that (1) $\vartheta$ achieves the maximum in Eq.(3) and (2) the corresponding path of $\vartheta$ in $\mathcal{G}_{un}$ visits nodes at every time step, then we have

$$d_r = \max_{i \in \mathbb{Z}} \left( \limsup_{n \to +\infty} \frac{\mathfrak{D}_i(n)}{n} \right) = \lim_{n \to +\infty} \frac{\mathfrak{D}_i(n)}{n}$$

*Proof.* We only need to prove, in such a graph, for all $i \in \mathbb{Z}$ we have

$$\liminf_{n \to +\infty} \frac{\mathfrak{D}_i(n)}{n} \ge \max_{i \in \mathbb{Z}} \left( \limsup_{n \to +\infty} \frac{\mathfrak{D}_i(n)}{n} \right) = d_r$$

Because it is obvious that

$$\mathrm{liminf}_{n \to +\infty} \frac{\mathfrak{D}_i(n)}{n} \le d_r$$

Namely, it suffice to prove, for all $i \in \mathbb{Z}$, for all $\epsilon > 0$, there is an $N_\epsilon > 0$, such that when $n > N_\epsilon$, we have $\frac{\mathfrak{D}_i(n)}{n} \ge d_r - \epsilon$. On the other hand, for $k \in \mathbb{N}$, if we assume $(k+1)\sigma_s(\vartheta) + i > n \ge i + k \cdot \sigma_s(\vartheta)$, then according to condition (2) we have

$$\frac{\mathfrak{D}_i(n)}{n} \ge \frac{k \cdot l(\vartheta)}{(k+1)\sigma_s(\vartheta)} = \frac{l(\vartheta)}{\sigma_s(\vartheta)} - \frac{l(\vartheta)}{\sigma_s(\vartheta)} \frac{1}{k+1}$$

We can see that if we set $k > \frac{\sigma_s(\vartheta)}{l(\vartheta)\epsilon}$, the inequality we wanted to prove.

$\square$

We now prove Proposition 3.3.1 and Theorem 3.4 as follows.

**Proposition A.1.2.** Given an RNN with recurrent depth $d_r$, we denote

$$d_f = \sup_{i, n \in \mathbb{Z}} \mathfrak{D}_i^*(n) - n \cdot d_r.$$

The supremum $d_f$ exists and we have the following least upper bound:

$$\mathfrak{D}_i^*(n) \leq n \cdot d_r + d_f.$$

*Proof.* We first prove that $d_f < +\infty$. Write $d_f(i) = \sup_{n \in \mathbb{Z}} \mathfrak{D}_i^*(n) - n \cdot d_r$. It is easy to verify $d_f(\cdot)$ is $m-$periodic, so it suffices to prove for each $i \in \mathbb{N}$, $d_f(i) < +\infty$. Hence it suffices to prove

$$\limsup_{n \to \infty}(\mathfrak{D}_i^*(n) - n \cdot d_r) < +\infty.$$

From the definition, we have $\mathfrak{D}_i(n) \geq \mathfrak{D}_i^*(n)$. So we have

$$\mathfrak{D}_i^*(n) - n \cdot d_r \leq \mathfrak{D}_i(n) - n \cdot d_r.$$

From the proof of Theorem A.1, there exists two constants $M'$ and $\Gamma$ depending only on the RNN $\mathcal{G}_c$, such that

$$\frac{\mathfrak{D}_i(n)}{n} \leq d_r + \frac{M' + \Gamma}{n}.$$

So we have

$$\limsup_{n \to \infty}(\mathfrak{D}_i^*(n) - n \cdot d_r) \leq \limsup_{n \to \infty}(\mathfrak{D}_i(n) - n \cdot d_r) \leq M' + \Gamma.$$

Also, we have $d_f = \sup_{i, n \in \mathbb{Z}} \mathfrak{D}_i^*(n) - n \cdot d_r$, so for any $i, n \in \mathbb{Z}$,

$$d_f \geq \mathfrak{D}_i^*(n) - n \cdot d_r.$$

$\square$

**Theorem A.2.** *Given an RNN and its two graph representations $\mathcal{G}_{\mathrm{un}}$ and $\mathcal{G}_c$, we denote $\xi(\mathcal{G}_c)$ the set of directed path that starts at an input node and ends at an output node in $\mathcal{G}_c$. For $\gamma \in \xi(\mathcal{G}_c)$, denote $l(\gamma)$ the length and $\sigma_s(\gamma)$ the sum of $\sigma$ along $\gamma$. Then we have:*

$$d_f = \sup_{i, n \in \mathbb{Z}} \mathfrak{D}_i^*(n) - n \cdot d_r = \max_{\gamma \in \xi(\mathcal{G}_c)} l(\gamma) - \sigma_s(\gamma) \cdot d_r.$$

*Proof.* Let $\gamma : \{(t_0, 0), (t_1, p_1), \cdots, (t_{n_\gamma}, p)\}$ be a path in $\mathcal{G}_{\mathrm{un}}$ from an input node $(t_0, 0)$ to an output node $(t_{n_\gamma}, p)$, where $t_0 = i$ and $t_{n_\gamma} = i + n$. We denote $\bar{\gamma}$ as the image of $\gamma$ on the cyclic graph. From the proof of Theorem A.1, for each $\bar{\gamma}$ in $\mathcal{G}_c$, we can decompose it into a path $\gamma_0$ and a sequence of directed cycles $C = C_1(\gamma), C_2(\gamma), \cdots, C_w(\gamma)$ on $\mathcal{G}_c$ satisfying those properties listed in Theorem A.1. We denote $l_0, l_1, \cdots, l_w$ to be the length of path $\gamma_0$ and directed cycles $C_1(\gamma), C_2(\gamma), \cdots, C_w(\gamma)$. We know $\frac{l_k}{\sigma_s(C_k)} \leq d_r$ for all $k = 1, 2, \ldots, w$ by definition. Thus,

$$l_k \leq d_r \cdot \sigma_s(C_k)$$

$$\sum_{k=1}^{w} l_k \leq d_r \cdot \sum_{k=1}^{w} \sigma_s(C_k)$$

Note that $n = \sigma_s(\gamma_0) + \sum_{k=1}^{w} \sigma_s(C_k)$. Therefore,

$$l(\gamma) - n \cdot d_r = l_0 + \sum_{k=1}^{w} l_k - n \cdot d_r$$

$$\leq l_0 + d_r \cdot (\sum_{k=1}^{w} \sigma_s(C_k) - n)$$

$$= l_0 - d_r \cdot \sigma_s(\gamma_0)$$

for all time step $i$ and all integer $n$. The above inequality suggests that in order to take the supremum over all paths in $\mathcal{G}_{\mathrm{un}}$, it suffices to take the maximum over a directed path in $\mathcal{G}_c$. On the other hand, the equality can be achieved simply by choosing the corresponding path of $\gamma_0$ in $\mathcal{G}_{\mathrm{un}}$. The desired conclusion then follows immediately.

$\square$

Lastly, we show Theorem 3.6.

**Theorem A.3.** *Given an RNN cyclic graph and its unfolded representation $(\mathcal{G}_c, \mathcal{G}_{un})$, we denote $C(\mathcal{G}_c)$ the set of directed cycles in $\mathcal{G}_c$. For $\vartheta \in C(\mathcal{G}_c)$, denote $l(\vartheta)$ the length of $\vartheta$ and $\sigma_s(\vartheta)$ the sum of $\sigma$ along $\vartheta$. Write $s_i = \liminf_{k \to \infty} \frac{\mathfrak{d}_i(n)}{n}$. We have :*

- *The quantity $s_i$ is periodic, in the sense that $s_{i+m} = s_i, \forall i \in \mathbb{N}$.*

- *Let $s = \min_i s_i$, then*

$$d_r = \min_{\vartheta \in C(\mathcal{G}_c)} \frac{l(\vartheta)}{\sigma_s(\vartheta)}. \tag{4}$$

*Proof.* The proof is essentially the same as the proof of the first theorem. So we omit it here. □

**Proposition A.3.1.** Given an RNN and its two graph representations $\mathcal{G}_{un}$ and $\mathcal{G}_c$, if $\exists \vartheta \in C(\mathcal{G}_c)$ such that $(1)$ $\vartheta$ achieves the minimum in Eq.(4) and $(2)$ the corresponding path of $\vartheta$ in $\mathcal{G}_{un}$ visits nodes at every time step, then we have

$$s = \min_{i \in \mathbb{Z}} \left( \liminf_{n \to +\infty} \frac{\mathfrak{d}_i(n)}{n} \right) = \lim_{n \to +\infty} \frac{\mathfrak{d}_i(n)}{n}.$$

*Proof.* The proof is essentially the same as the proof of the Proposition A.1.1. So we omit it here. □

# B    Experiment Details

## B.1    RNNs with tanh

In this section we explain the functional dependency among nodes in RNNs with tanh in detail.

The transition function for each node is the tanh function. The output of a node $v$ is a vector $h_v$. To compute the output for a node, we simply take all incoming nodes as input, and sum over their affine transformations and then apply the tanh function (we omit the bias term for simplicity).

$$h_v = \tanh\left(\sum_{u \in \text{In}(v)} \mathbf{W}(u)h_u\right),$$

where $\mathbf{W}(\cdot)$ represents a real matrix.

Figure 3: "Bottom-up" architecture ($bu$).

As a more concrete example, consider the "bottom-up" architecture in Figure 3, with which we did the experiment described in Section 4.2. To compute the output of node $v$,

$$h_v = \tanh(\mathbf{W}(u)h_u + \mathbf{W}(p)h_p + \mathbf{W}(q)h_q). \tag{5}$$

## B.2    LSTMs

In this section we explain the Multidimensional LSTM (introduced by [1]) which we use for experiments with LSTMs.

The output of a node $v$ of the LSTM is a 2-tuple ($c_v$, $h_v$), consisting of a cell memory state $c_v$ and a hidden state $h_v$. The transition function $F$ is applied to each node indistinguishably. We describe the computation of $F$ below in a sequential manner (we omit the bias term for simplicity).

$$z = g\left(\sum_{u \in \text{In}(v)} \mathbf{W}_z(u)h_u\right) \qquad\qquad \text{block input}$$

$$i = \sigma\left(\sum_{u \in \text{In}(v)} \mathbf{W}_i(u)h_u\right) \qquad\qquad \text{input gate}$$

$$o = \sigma\left(\sum_{u \in \text{In}(v)} \mathbf{W}_o(u)h_u\right) \qquad\qquad \text{output gate}$$

$$\{f_u\} = \left\{\sigma\left(\sum_{u' \in \text{In}(v)} \mathbf{W}_{f_u}(u')h_u\right) | u \in \text{In}(v)\right\} \qquad \text{A set of forget gates}$$

$$c_v = i \odot z + \sum_{u \in \text{In}(v)} f_u \odot c_u \qquad\qquad \text{cell state}$$

$$h_v = o \odot c_v \qquad\qquad \text{hidden state}$$

Note that the Multidimensional LSTM includes the usual definition of LSTM as a special case, where the extra forget gates are 0 (i.e., bias term set to -$\infty$) and extra weight matrices are 0. We again consider the architecture $bu$ in Fig. 3. We first compute the block input, the input gate and the output

gate by summing over all affine transformed outputs of $u, p, q$, and then apply the activation function. For example, to compute the input gate, we have

$$i = \sigma\left(\mathbf{W}_i(u)h_u + \mathbf{W}_i(p)h_p + \mathbf{W}_i(q)h_q\right).$$

Next, we compute one forget gate for each pair of $(v, u), (v, p), (v, q)$. The way of computing a forget gate is the same as computing the other gates. For example, the forget gate in charge of the connection of $u \to v$ is computed as,

$$f_u = \sigma\left(\mathbf{W}_{f_u}(u)h_u + \mathbf{W}_{f_u}(p)h_u + \mathbf{W}_{f_u}(q)h_u\right).$$

Then, the cell state is simply the sum of all element-wise products of the input gate with the block output and forget gates with the incoming nodes' cell memory states,

$$c_v = i \odot z + f_u \odot c_u + f_p \odot c_p + f_q \odot c_q.$$

Lastly, the hidden state is computed as usual,

$$h_v = o \odot c_v.$$

### B.3 Recurrent Depth is Non-trivial

The validation curves of the 4 different connecting architectures $sh$, $st$, $bu$ and $td$ on text8 dataset for both $\mathrm{tanhRNN}$-small and LSTM-small are shown below:

Figure 4: Validation curves for $sh$, $st$, $bu$ and $td$ on test8 dataset. Left: results for $\mathrm{tanhRNN}$-small. Right: results for LSTM-small.

### B.4 Full Comparisons on Depths

Figure 5 shows all the validation curves for the 9 architectures on text8 dataset, with their $d_r = 1, 2, 3$ and $d_f = 2, 3, 4$ respectively. We initialize hidden-to-hidden matrices from uniform distribution.

Also, to see if increasing feedforward depth/ recurrent depth helps for long term dependency problems, we evaluate these 9 architectures on sequential MNIST task, with roughly the same number of parameters( 8K, where the first architecture with $d_r = 1$ and $d_f = 2$ has hidden size of 90.). Hidden-to-hidden matrices are initialized from uniform distribution.

Figure 6 clearly show that, as the feedforward depth increases, the model performance stays roughly the same. In addition, note that increasing recurrent depth might even result in performance decrease. This is possibly because that larger recurrent depth amplifies the gradient vanishing/exploding problems, which is detrimental on long term dependency tasks.

Figure 5: Validation curves of 9 architectures with feedforward depth $d_f = 2, 3, 4$ and recurrent depth $d_r = 1, 2, 3$ on test8 dataset. For each figure in the first row, we fix $d_f$ and draw 3 curves with different $d_r = 1, 2, 3$. For each figure in the second row, we fix $d_r$ and draw 3 curves with different $d_f = 2, 3, 4$.

Figure 6: Test accuracies of 9 architectures with feedforward depth $d_f = 2, 3, 4$ and recurrent depth $d_r = 1, 2, 3$ on sequential MNIST. For each figure, we fix $d_r$ and draw 3 curves with different $d_f$.

## B.5   Recurrent Skip Coefficients

The test curves for all the experiments are shown in Figure 7. In Figure 7, we observed that obtaining good performance on MNIST requires larger $s$ than for pMNIST. We hypothesize that this is because, for the sequential MNIST dataset, each training example contains many consecutive zero-valued subsequences, each of length 10 to 20. Thus within those subsequences, the input-output gradient flow could tend to vanish. However, when the recurrent skip coefficient is large enough to cover those zero-valued subsequences, the model starts to perform better. With $p$MNIST, even though the random permuted order seems harder to learn, the permutation on the other hand blends zeros and ones to form more uniform sequences, and this may explain why training is easier, less hampered by by the long sequences of zeros.

## B.6   Recurrent Skip Coefficients vs. Skip Connections

Test curves for all the experiments are shown in Figure 8. Observe that in most cases, the test accuracy of (3) is worse than (2) in the beginning while beating (2) in the middle of the training. This is possibly because in the first several time steps, it is easier for (2) to pass information to the output thanks to the skip connections, while only after multiples of $k$ time steps, (3) starts to show its advantage with recurrent skip connections[11]. The shorter paths in (2) make its gradient flow

more easily in the beginning, but in the long run, (3) seems to be more superior, because of its more prominent skipping effect over time.

Figure 7: Test curves on MNIST/$p$MNIST, with `tanh` and $LSTM$. The numbers in the legend denote the recurrent skip coefficient $s$ of each architecture.

Figure 8: Test curves on MNIST/$p$MNIST for architecture (1), (2), (3) and (4), with `tanh`. The recurrent skip coefficient $s$ of each architecture is shown in the legend.

## Footnotes

[10] $\mathfrak{D}_i(n)$ is not defined when there does not exist a path from time $i$ to time $i + n$. We simply omit undefined cases when we consider the limsup. In a more rigorous sense, it is the limsup of a subsequence of $\{\mathfrak{D}_i(n)\}_{n=1}^{\infty}$, where $\mathfrak{D}_i(n)$ is defined.

[11] It will be more clear if one checks the length of the shortest path from an node at time $t$ to to a node at time $t + k$ in both architectures.

# References

[1] Alex Graves, Santiago Fernández, and Jürgen Schmidhuber. Multi-dimensional recurrent neural networks. In *Proceedings of the 17th International Conference on Artificial Neural Networks*, ICANN'07, pages 549–558, Berlin, Heidelberg, 2007. Springer-Verlag.