[Reviews · NeurIPS 2016]

Reviewer 1

Summary

This paper proposes several definitions of measures of complexity of a recurrent neural network. They measure 1) recurrent depth (degree of multi-layeredness as a function of time of recursive connections) 2) feedforward depth (degree of multi-layeredness as a function of input -> output connections) 3) recurrent skip coefficient (degree of directness, like the inverse of multilayeredness, of connections) In addition to the actual definitions, there are two main contributions: - The authors show that the measures (which are limits as the number of time steps -> infinity) are well defined. - The authors correlate the measures with empirical performance in various ways, showing that all measure of depth can lead to improved performance.

Qualitative Assessment

The measures proposed by the paper seem intuitively sensible and useful. One could argue importance of these measures is not clear if they are simply demonstrated (non entirely conclusively as it turns out) by comparison with empirical results on real data. In other words, if some formal guarantees could be provided as a function of these measures, rather than empirical evaluations, the usefulness of the measures would be undeniable. At present, one could ask why these measures are more relevant than other possible measures of complexity. It is also a little unfortunate that so much complexity is left in the appendix, but this is a common trend in nips and the authors can’t be blamed for that.

Confidence in this Review

2-Confident (read it all; understood it all reasonably well)


Reviewer 2

Summary

This paper provides 3 measures of complexity for RNNs. They then show experimentally that these complexity measures are meaningful, in the sense that increasingly complexity seems to correlated with better performance.

Qualitative Assessment

Update: In line 71, there is a V without a subscript. I think it should be V_c. --- I like this paper, but some figures need better explanation. Figure 1(a) is quite clear, as it corresponds to definitions 2.1 and 2.2. However, Figure 1(b) and Figure 2 are using a different notation and it took me quite some time to decode. The caption for Figure 2 "We only plot the hidden states within 1 time step (which also have a period of 1)" is too cryptic to be very helpful. In particular arrows now indicate nonlinear transformations, including implicit input/output nodes. It's particularly confusing in 2a because, without only 2 time slices, it looks like the cyclic representation rather than the unrolled representation (except for the lack of cycles). Next, and I guess you probably won't want to change this, but it seems like your definitions and explanations get significantly heavier and more complicated to allow for m > 1, with very little indication of benefit: If we only allowed RNNs with m=1, would the set of RNNs actually change? Or does the use of m > 1 just make explicit one particular type of structure an RNN can have? If so, what's so special about the period number that we want to call that out in particular? In any case, a sentence or two of discussion to address these questions would be beneficial, I think. Small notes: -In the definition of unfolding, in the definition of V_{un}, I think you want a subscript of c on the V? - In the definitions, don't we want to prevent the weight sigma from being negative?

Confidence in this Review

2-Confident (read it all; understood it all reasonably well)


Reviewer 3

Summary

This is a good theory paper! The authors first present a rigorous graph-theoretic framework that describes the connecting architectures of RNNs in general, with which the authors easily explain how we can unfold an RNN. The authors then go on and propose tree architecture complexity measures of RNNs, namely the recurrent depth, the feedforward depth and the recurrent skip coefficient. Experiments on various tasks show the importance of certain measures on certain tasks, which indicates that those three complexity measures might be good guidelines when designing a recurrent neural network for certain tasks.

Qualitative Assessment

1. On line 59, shouldn't it be $i \in {0, 1, 2, ..., m-1}$? 2. In the caption of Figure 1, it will be better if you add brief description of how you compute $d_r$, $d_f$ and $s$ for networks in (b). 3. On line 155, I think it should be $d_f = sup_{i, n \in \mathbb{Z}}(\mathfrak(D)_i^{\ast}(n) - n \cdot d_r)$, i.e., you have to add "()" for sup? 4. On line 167, same as above, do you have to add "()" for sup and max? 5. I notice you conduct experiments on NLP tasks, image classification tasks. I think it will be a good idea if you add experiments for speech recognition tasks as well, as recurrent neural networks are extensively used in that field. This will also bring the paper to broader potential readers. Adding speech recognition results should be easy nowadays. E.g., you can search for Kaldi, Librispeech, etc. They have existing recipes for different neural networks and they even have results. You just have to do the analysis.

Confidence in this Review

2-Confident (read it all; understood it all reasonably well)


Reviewer 4

Summary

Three complexity measures of recurrent neural networks is proposed for recurrent neural networks. 1. recurrent depth; 2. feedforward depth; 3. recurrent skip coefficients. Experimental results on parsing, language modeling, program modeling, and image modeling evaluates the importance of these complexity measures on these applications.

Qualitative Assessment

I think this paper provides a useful empirical evaluation on the effects of different complexity measures. The evaluation is very useful to the community.

Confidence in this Review

2-Confident (read it all; understood it all reasonably well)


Reviewer 5

Summary

This paper provides an analysis of the Recurrent Neural Net (RNN) architecture by defining several measurements: 1). Recurrent depth, d_r; 2). Feedforward depth, d_f; 3). Skip coefficient s. In the experiment, the paper compares several RNN variants with different d_r, d_f, and s on five standard datasets and problems. Based on these experiments, this paper offers some guidelines about how to design an RNN.

Qualitative Assessment

The goal of the paper is to provide a theoretical and quantitative guideline for designing an RNN for a specific problem. This is an important and hard topic that worth investigating. The measurements of d_r (Recurrent depth), d_f (Feedforward depth) and s (Skip coefficient) seem to be reasonable. The authors also conduct comprehensive experiments to compare several RNN variants. However, I find the paper hard to understand. The idea behind the proposed measurements is actually very simple, but the paper uses too much symbols and definitions to explain it. In many cases, the symbols just appear without being explained or defined before (e.g. “s” in the caption of Figure 1 in Page 3 is defined in Page 5.). The reader sometimes needs to manually search for a specific symbol. I strongly recommend the authors to remove unnecessary symbols and rewrite the paper using plain language as best as they can. It would be also very helpful if the author could summarize the meaning of the important symbols in a table at the beginning of the paper. Several definitions and lemmas can also be moved to the supplementary material. This will significantly improve the clearance and impact of the paper. I also have several questions about the experiments: 1. In Table 1, the performance of different models seems to be too close to each other. E.g. (1.84 v.s. 1.83). What is the performance variance of the same model with different random initializations? If the experiment is conducted again, will the conclusion be the opposite to the current one? Also, although BPC in the caption has been explained before in text, it would be better if you write Bit-Per-Character at least once in the caption since the reader might not be very familiar with this term. 2. In Table 2, in the top-left table the best performance of MNIST is 87.8. But when you try to compare your model with other papers, the best performance is changed to 98.1 shown in the bottom-left table. It is really confusing to me. I guess the models are different. In the top left table the model is RNN(tanh). In the bottom left table the model is RNN(stanh)? But why not just compare RNN(stanh) models with different s in the top left table? Please explain. 3. Minor: Please add a vertical line in Figure 2 to separate the right and left sub-figures. In summary, this is an interesting paper studying an important problem. However, the current form of the paper is unclear and sometimes confusing. It needs more polishing in order to be accepted in NIPS.

Confidence in this Review

2-Confident (read it all; understood it all reasonably well)


Reviewer 6

Summary

In this work the authors define three different complexity measures for RNNs: recurrent depth, feedforward depth and the recurrent skip coefficient. They give definitions for these three measures within a precise graph-theoretic framework. They then present experiments on various sequential tasks for RNN models with different complexity according to these measures.

Qualitative Assessment

A significant portion of the paper is spent defining in a very rigorous way three complexity measures which are fairly obvious, and in my opinion do not need the heavy graph-theoretic machinery the authors make use of. For example, the authors give intuitive and clear definitions of recurrent depth and skip coefficient (respectively, as the average maximum number of non-linear transformations per time step, and the length of the shortest path from time i to time (i+n)). These are well illustrated with Figure 1 and the formal definitions the authors give in terms of the graph-theoretic framework do not add anything fundamental. Formalizing everything in a rigorous framework could be useful if the authors were proving new and useful theorems, but the theorems they provide are limited to proving the computability of the the measures they define, which seems more of a technical detail. It is not clear what conclusions to draw from the experiments. The investigation of the different architectures in Figure 2a and 2b seems like an interesting direction, but the experiments are not very systematic and there are some methodological issues. Questions/comments: -Why do the authors evaluate the architectures in Figure 2a on PennTree and Text8, the architectures in Figure 2b on Text8 and MNIST? Wouldn't it make more sense to evaluate the same set of architectures on all the datasets and see if certain architectures consistently work better on certain types of problems? -are the models whose results are in Table 1 (right) the same size as those reported in Table 1 (left) for Text8? It is not clear if the difference in performance is due to the differences in architecture or model size. - It is not clear if the improvements with recurrent depth reported in Table 1 (right) are statistically significant, error bars over several initializations would help. - In Section 4.3, the results for MNIST should be included in the main paper, not the appendix. Also, they should be presented in a table like Table 1 (right). - In Section 4.4, the results on the adding and copying problem should be presented in a table, not the text. - In Table 2, top left: it is surprising that the results for permuted MNIST are *better* than the unpermuted version, since presumably this increases the timescale of the dependencies. Do you have an explanation? - Sentences like line 294-296: "We varied...found that this kind of improvement persists" should refer to specific results. It isn't clear what insights the experiments give. The authors say in the conclusion "we find empirical evidence that increasing recurrent depth might yield performance improvements", yet increasing recurrent depth does not help for sequential MNIST and it is not tried for the copy or addition task. They also say "increasing feedforward depth might not help on long term dependency tasks", which is unsurprising since it should not have a big influence on vanishing or exploding gradients. Their third point is that "increasing recurrent skip coefficient can largely improve performance on long-term dependency task", which was already known (for example, the Clockwork RNN).

Confidence in this Review

2-Confident (read it all; understood it all reasonably well)